# Activating the Intrinsic Pathway of Apoptosis Using BIM BH3 Peptides Delivered by Peptide Amphiphiles with Endosomal Release

**DOI:** 10.3390/ma12162567

**Published:** 2019-08-12

**Authors:** Mathew R. Schnorenberg, Joseph A. Bellairs, Ravand Samaeekia, Handan Acar, Matthew V. Tirrell, James L. LaBelle

**Affiliations:** 1Pritzker School of Molecular Engineering, University of Chicago, Chicago, IL 60637, USA; 2Department of Pediatrics, Section of Hematology/Oncology, University of Chicago, Chicago, IL 60637, USA; 3Medical Scientist Training Program, University of Chicago, Chicago, IL 60637, USA

**Keywords:** peptide amphiphile, peptide therapeutics, cathepsin, BCL-2 family, BIM, BH3-only, apoptosis

## Abstract

Therapeutic manipulation of the BCL-2 family using BH3 mimetics is an emerging paradigm in cancer treatment and immune modulation. For example, peptides mimicking the BIM BH3 helix can directly target the full complement of anti- and pro-apoptotic BCL-2 proteins to trigger apoptosis. This study has incorporated the potent BH3 α-helical death domain of BIM into peptide amphiphile (PA) nanostructures designed to facilitate cellular uptake and induce cell death. This study shows that these PA nanostructures are quickly incorporated into cells, are able to specifically bind BCL-2 proteins, are stable at physiologic temperatures and pH, and induce dose-dependent apoptosis in cells. The incorporation of a cathepsin B cleavable linker between the BIM BH3 peptide and the hydrophobic tail resulted in increased intracellular accumulation and mitochondrial co-localization of the BIM BH3 peptide while also improving BCL-2 family member binding and apoptotic reactivation. This PA platform represents a promising new strategy for intracellular therapeutic peptide delivery for the disruption of intracellular protein:protein interactions.

## 1. Introduction

The BCL-2 family of proteins forms a complex protein-protein interaction (PPI) network that regulates cellular life and death decisions, which contributes to organismal development, cancer ontogeny and chemoresistance, hematopoiesis, and immune regulation [1,2,3,4]. Members of the BCL-2 family known as BH3-only proteins (e.g., BIM, BID, PUMA, NOXA) serve as cellular stress sentinels and can trigger irreversible activation of apoptosis through their α-helical BH3 death domains. These pro-apoptotic signals are normally held in check by the multidomain anti-apoptotic family members (e.g., BCL-XL, BCL-2, BCL-W, MCL-1) through sequestering PPIs. However, when pro-apoptotic signals outweigh anti-apoptotic signals, the multidomain pro-apoptotic effectors, BAX and BAK, are activated and trigger the cell death cascade by oligomerizing in the mitochondrial membrane, leading to mitochondrial outer membrane permeabilization (MOMP), cytochrome c release, apoptosome formation, and effector caspase activation [1].

The BCL-2 family’s PPIs have emerged as an impactful set of therapeutic targets [5]. Cancer cells often push the BCL-2 family’s PPI balance toward an anti-apoptotic state to avoid cell death despite cellular stress and damage, for example by upregulating anti-apoptotic members or downregulating pro-apoptotic members [5,6]. As most chemotherapies function by ultimately inducing apoptosis, cancers often acquire chemotherapeutic resistance by manipulating the homeostatic balance of BCL-2 family members [6]. BH3-mimetics are a powerful way to therapeutically interrupt this balance and reactivate cell death, particularly in cancers that are “primed for death” with upregulated anti-apoptotic proteins [7].

While intracellular PPIs are commonly deemed undruggable targets, peptides can effectively mimick the PPI interface domains of proteins and thus disrupt PPIs with high specificity and affinity. However, because peptides are typically not cell permeable, chemical poration of the cell membrane is required for peptides to reach their intracellular targets, making many potential peptides irrelevant for therapeutic applications. Hydrocarbon-stapled peptides are a noteworthy exception in which an all-hydrocarbon staple is installed across helical turns of a peptide. These stapled peptides have enhanced α-helicity, resist proteolysis, and, with empirical screening, can be made cell-permeable [8,9,10].

Recent work has shown that peptide amphiphiles (PAs), or peptides linked to a hydrophobic, lipid-like tail, can also impart cellular uptake for otherwise cell-impermeable peptides [11,12,13]. Moreover, these PAs can spontaneously self-assemble into micellar nanostructures in aqueous solution. Supramolecular peptide delivery provides several advantages compared to peptides alone: single micelles can (1) deliver high concentrations of peptides into cells; (2) stabilize peptide secondary structure(s); (3) protect peptides from proteolysis in the blood stream; (4) increase circulation half-lives; and (5) simultaneously deliver multiple therapeutics targeting non-redundant, synergistic cellular pathways [14]. We have also previously shown that PA micelles can be actively targeted to specific cell types in vivo, where a targeting PA can successfully carry non-targeted cargo to a target receptor simply through their supramolecular co-assembly [15,16]. 

While the exact mechanism of PA cellular uptake remains unproven, recent work has shown that PAs interact with the cell membrane and traffic through endosomes and lysosomes [11,12,13,17]. However, facile intracellular delivery and the release from endosomal/lysosomal compartments have been significant limitations to using PAs for intracellular delivery of biofunctional peptides. We recently showed that incorporating an endosomally-cleavable linker between the peptide cargo and lipid tail enhances intracellular accumulation and minimizes lipid-driven recycling out of the cell [18]. 

Here, this study develops and biochemically characterizes novel PAs able to intracellularly deliver an α-helical peptide mimicking the BH3 death domain of BIM previously shown to bind BCL-2 family members and potently reactivate apoptosis in resistant malignancies [19,20,21,22]. In so doing, these PAs overcome membrane sequestration, plasma membrane recycling, and lysosomal degradation limitations known to decrease the potency of first-generation PAs. Using native peptides, which by themselves are cell-impermeable, the BIM BH3 domain was attached to hydrophobic tails and thereby incorporated into spherical micelles, which are ideal for in vivo delivery and trafficking [23]. Lastly, intracellular organelle sequestration of the biofunctional peptide was overcome through the incorporation of a cathepsin-cleavable linker between the peptide and hydrophobic tail.

## 2. Materials and Methods

### 2.1. Micelle Synthesis

Two overlapping but functionally non-equivalent sequences of the BCL-2 binding motif of the BIM protein (BIM_A_: IWIAQELRRIGDEFNAYYARR [19], and BIM_B_: MRPEIWIAQELRRIGDEFNA [24,25]) were synthesized on 0.25 mmoles of rink amide resin (Novabiochem) through standard Fmoc-mediated solid phase peptide synthesis methods using an automated PS3 Benchtop Peptide Synthesizer (Protein Technologies, Tucson, AZ, USA). Amino acids were used in 4X excess. Each coupling began with Fmoc-deprotection of the resin using 20% piperidine in dimethylformamide (DMF), followed by amino acid activation with N,N,N′,N′-Tetramethyl-O-(1H-benzotriazol-1-yl)uronium hexafluorophosphate (HBTU) and 0.4 M N-methylmorpholine in DMF before adding the activated amino acid to the resin. Five lysines were added at the C-termini of both sequences to make them more charged and water-soluble. After splitting each batch of the peptides in half, one half was acetylated with acetic anhydride in DMF, while the other half was labeled with fluorescein isothiocyanate (FITC; Molecular Probes) at their N-termini. The acetylated and FITC-labeled peptides were again each divided in half. One half was conjugated at the ε-amine of C-terminal lysine to dipalmitoylglutamic acid, or diC16, as described previously [18,26] to form Ac-BIM-PA_1_ or FITC-BIM-PA_1_. The other half was coupled with 1,2-distearoyl-sn-glycero-3-phosphoethanolamine-N-[succinyl(polyethylene glycol)-2000], or DSPE-PEG(2000)-succinyl (Avanti Polar Lipids, Alabaster, AL, USA), using equal molar equivalent of peptide to lipid in a 1:1 mixture of DMF:DCM and named Ac-BIM-PA_2_ or FITC-BIM-PA_2_. The same peptides were re-synthesized with a cathepsin cleavage sequence—Valine, Citrulline (VCit)—incorporated between the BIM sequences and the five C-terminal lysines, and the modified PAs were named Ac-BIM_cath,K_PA_2_ or FITC-BIM_cath,K_PA_2_. Peptide amphiphiles were then cleaved with 82.5:5:5:2.5 by volume trifluoroacetic acid: H2O: phenol: thioanisole: 1,2-ethanedithiol. They were precipitated and washed with cold diethyl ether, dried under nitrogen, and dissolved in water. Reverse-phase HPLC (Prominence, Shimadzu, Columbia, MD, USA) on a C8 column (Waters, Milford, MA, USA) at 40 °C was employed to purify soluble peptides using a gradient method and acetonitrile/water solvents containing 0.1% formic acid. The fractions were mass-characterized by MALDI-TOF mass spectral analysis (Biflex III, Bruker, Billerica, MA, USA) and the confirmed fractions were aliquoted, lyophilized, and stored as powders at −20 °C (Appendix A). The exact concentrations of the aliquots were determined by amino acid analysis (AAA). To fabricate BIM micelles, the lyophilized powders were reconstituted, sonicated for 1 h at room temperature, incubated in 70 °C water bath for 1 h, and left at room temperature for at least 2 h to cool down and equilibrate.

### 2.2. Critical Micelle Concentration (CMC)

To measure the CMC of PA micelles, 1,6-diphenyl-1,3,5-hexatriene (DPH) was first dissolved in tetrahydrofuran and then diluted in water to 1 μM final concentration. A range of PA solutions (from 512 μM to 0.01 μM in serial one-half dilutions) were prepared in a 1 μM DPH solution and left to equilibrate for 1 h at room temperature. These solutions were plated in triplicates in a 96-well plate, and their fluorescence intensity was measured using a Tecan Infinite 200 plate reader (Mannedorf, Switzerland). DPH was excited at 350 nm, and fluorescence emission was measured at 428 nm. The concentration values were log transformed, and the data were fit with a trend line for the increasing intensity measurements. The CMC relies on the partitioning of DPH from the aqueous solution into the hydrophobic core of intact PA micelles, resulting in a sharp increase in fluorescence intensity. The CMC was calculated as the inflection point where the fluorescence intensity began to increase. 

### 2.3. Dynamic Light Scattering (DLS)

Stock solutions of 0.5 mM BIM PAs were prepared as mentioned above, and DLS measurements were performed at a 90° angle with a 637 nm laser using a Brookhaven Instruments (Holtzville, NY, USA) system with a BI-9000AT autocorrelator. Hydrodynamic radii were determined via the Stokes-Einstein equation using the diffusion coefficient determined from the auto correlation function.

### 2.4. Transmission Electron Microscopy (TEM)

Ultrathin carbon type-A 400 mesh copper grids (Ted Pella, Redding, CA, USA) were loaded with 1 μL of BIM PAs and allowed to dry. The grids were washed with several drops of water and then negatively stained with 1% aqueous phosphotungstic acid for 1 min. The excess solution was then removed and grids were left to dry. All the grids were imaged on a FEI Tecnai 12 TEM (Hillsboro, OR, USA) using an accelerating voltage of 120 kV.

### 2.5. Circular Dichroism

A quartz cuvette with a 0.1 cm pathlength was loaded with 200 μL of 50 μM solutions of BIM PAs. Measurements were done at 25, 37, 42 and 50 °C with a Jasco J-815 Circular Dichroism Spectropolarimeter (Easton, MD, USA). Each sample was scanned five times from 250 to 190 nm, and the data were averaged. Mean residue ellipticity [θ] was calculated using Equation (1):[θ] = millidegree/molar concentration/amino acid residues,(1)
with units degree x cm^2^ x dmol^−1^ x residue^−1^. Percent alpha-helicity was then calculated from the value of [θ] at 222 nm using Equations (2) and (3):(2)$$\%\;Helicity\;=\;100\times {\left[\theta \right]}_{222}\;/\;[\theta ]{\;}_{222}^{max}\;\mathrm{where}$$
(3)$$[\theta ]{\;}_{222}^{max}=-40,000\times \left[1-\left(2.5/amino\;acid\;residues\right)\right]+100\times T$$
with *T* measured in °C [27,28].

### 2.6. Lactate Dehydrogenase Release Assay

Mouse embryonic fibroblasts (MEFs) were cultured in a 96-well plate (5000 cells per well) and allowed to adhere overnight. A serial dilution of the BIM PAs (25, 12.5, 6.25, 3.125, and 1.563 μM) as well as 1% triton were prepared in treatment media (Advanced DMEM, 1% FBS, 0.1% penicillin/streptomycin), and the cells’ media was replaced with the treatment dilutions to a final volume of 100 μL in each well. After incubation at 37 °C for 1 h, 50 μL of the treatment media was removed carefully from each well, transferred to a new plate, and mixed with 50 μL of lactate dehydrogenase (LDH) reagent (Roche) for 30 min while shaking, and absorbance was then measured at 492 nm on a microplate reader (SpectraMax M5 Microplate Reader, Molecular Devices, San Jose, CA, USA).

### 2.7. Protein Production

Recombinant and tagless BCL-XLΔC, MCL-1ΔNΔC, BCL-2ΔC and BCL-WΔC were produced and purified as described previously [19]. Briefly, glutathione-S-transferase fusion proteins were expressed in Escherichia coli BL21 (DE3) using pGEX2T (Pharmacia Biotech) constructs. Bacterial cells were cultured in ampicillin-containing Luria Broth, and protein expression was induced with 0.5 mM isopropyl β-D-1-thiogalactopyranoside. The bacterial pellet was resuspended in PBS containing 1 mg/mL lysozyme, SigmaFAST protease inhibitor tablet (Sigma-Aldrich, St. Louis, MO, USA), and 1% (*v*/*v*) Triton X-100. Bacteria were lysed by sonication at 4 °C, and, after centrifugation at 16,000× *g* for 30 min, the supernatant was applied to a glutathione-agarose (Sigma-Aldrich, St. Louis, MO, USA) column and washed with PBS. Tagless protein was obtained by overnight on-bead digestion with 50 units of thrombin (GE Healthcare, Pittsburgh, PA, USA) in 3 mL PBS at room temperature. The cleaved proteins were purified by size exclusion chromatography using 150 mM NaCl and 50 mM Tris (pH 7.4) buffer conditions on a Superdex-75 gel filtration column (GE Healthcare, Pittsburgh, PA, USA).

### 2.8. Fluorescence Polarization (FP) Binding Assay

FP binding assays were performed as previously described [19,29]. FITC-labeled peptides and peptide amphiphiles (50 nM) were incubated with serial dilutions of recombinant protein in FP binding buffer (50 mM Tris, 100 mM NaCl, pH 8.0) until equilibrium was reached. FP was measured using a SpectraMax M5 microplate reader (Molecular Devices, San Jose, CA, USA). To calculate K_d_ values, the data were fitted to normalized sigmoidal curves with a variable slope using nonlinear regression analysis in Graphpad Prism.

### 2.9. Cathepsin B Cleavage FP and Fluorogenic Assays

Cathepsin B purified from human liver (Sigma-Aldrich, St. Louis, MO, USA) was diluted 1:40 in cathepsin B activation buffer (25 mM HEPES, 25 mM DTT, pH 5.0) and incubated at room temperature for 15 min to activate the enzyme. FITC- BIM_A,cath,K_PA_2_ was added to activated enzyme to achieve a final peptide amphiphile concentration of 50 μM and incubated at room temperature for 30 min. The reaction was stopped by diluting the reaction mixture 1:100 in FP binding buffer (50 mM Tris, 100 mM NaCl, pH 8.0). The enzyme-treated peptide amphiphile (50 nM) was then incubated with serial dilutions of recombinant protein in FP binding buffer until equilibrium was reached. FP was measured using a SpectraMax M5 microplate reader (Molecular Devices, San Jose, CA, USA), and K_d_ values were calculated as described above.

The fluorogenic cathepsin-cleavage assay was done as previously described [18] using the cathepsin B cleavage substrate Z-Arg-Arg-7-amido-4-methylcoumarin hydrochloride (Z-RR-AMC; Sigma-Aldrich, St. Louis, MO, USA) and human cathepsin B with or without the cathepsin inhibitor CA-074Me (EMD Millipore, Burlington, MA, USA).

### 2.10. Cell Culture

Mouse embryonic fibroblasts (MEFs) and HeLa cells were maintained in DMEM (Invitrogen, Carlsbad, CA, USA) supplemented with 10% FBS, 100 U/mL penicillin/streptomycin, 2 mM L-glutamine, and 0.1 mM MEM nonessential amino acids. 

### 2.11. Live Cell Confocal Microscopy

HeLa cells were incubated with 10 μM FITC-labeled peptides or PAs for the indicated time in Advanced DMEM (Invitrogen, Carlsbad, CA, USA) supplemented with 1% FBS. The media was then removed, and the cells were washed and then incubated in prewarmed Opti-MEM (Invitrogen, Carlsbad, CA, USA) containing 250 nM MitoTracker Red (Invitrogen, Carlsbad, CA, USA) and 5 μg/mL Hoechst 33342 (Invitrogen, Carlsbad, CA, USA) for 30 min. The cells were then washed and incubated in Opti-MEM media lacking phenol red (Invitrogen, Carlsbad, CA, USA). Confocal images were collected on an Olympus DSU spinning disk confocal system (Olympus, Carlsbad, CA, USA) with a heated platform and a humidified chamber with 5% CO_2_. Excitation of the 3 fluorophores was performed sequentially using 405-nm, 488-nm, and 561-nm lasers. Images were acquired using a 100 Plan Apo objective lens with a Hamamatsu EM-CCD camera (Andor Technology, Belfast, UK). Acquisition parameters, shutters, filter positions, and focus were controlled by Slidebook 6 software (Intelligent Imaging Innovations, Denver, CO, USA).

### 2.12. Cell Viability Assay

MEF cells were aliquoted (2.5 × 10^3^, 100 μL) in 96-well opaque plates in complete DMEM media, and, 24 h later (at 75%–90% cellular confluence), the media was removed. The indicated doses of peptides or PAs were then added in Advanced DMEM (Invitrogen, Carlsbad, CA, USA) supplemented with 1% FBS, and after 6 h of treatment, 10% FBS was added back to the media. Cell viability was measured at the indicated time points by the addition of CellTiter-Glo chemiluminescence reagent in accordance with the manufacture’s protocol (Promega, Madison, WI, USA). Luminescence was detected by a Synergy 2 microplate reader (BioTek Instruments, Inc., Winooski, VT, USA).

### 2.13. Caspase-3/7 Activation Assay

Cells were treated as described above for the cell viability assays, and caspase-3/7 activation was measured at indicated time points by addition of the Caspase-Glo 3/7 chemiluminescence reagent in accordance with the manufacturer’s protocol (Promega, Madison, WI, USA). Luminescence was detected by a Synergy 2 microplate reader (BioTek Instruments, Inc., Winooski, VT, USA). Caspase-3/7 activation per living cells was determined by the ratio of Caspase-Glo luminescence to the percent viability from the corresponding CellTiter-Glo assay from identical experiments plated simultaneously, as previously described [30].

### 2.14. Western Blotting

Treated MEFs were collected and lysed in lysis buffer (50 mM Tris, 150 mM NaCl, 1 mM EDTA, 1 mM DTT, 1% [*v*/*v*] Triton X-100, complete protease inhibitor tablet [Roche], pH 7.4; PBST), and the protein content of each lysate was quantified using BCA kit (Thermo-Fischer Scientific, Waltham, MA, USA). Further, 5 μg of total protein from each lysate was loaded and separated on a 12% SDS-PAGE gel, transferred onto a PVDF membrane, and blocked with 5% skim milk in PBST for 45 min. The membranes were probed with primary antibody overnight at 4 °C with antibodies against PARP (Cell Signaling, Danvers, MA, USA; 1:1000) and GAPDH (Santa Cruz, Dallas, TX, USA; 1:1000), followed by 1 h of incubation at room temperature with HRP-conjugated secondary antibody (Santa Cruz, Dallas, TX, USA; 1:8000). Immuno-reactivity was visualized with a chemiluminescent detection kit (Amersham, Little Chalfont, UK).

### 2.15. Confocal Imaging after Cathepsin Inhibition

MEFs were cultured on coverslips inside 6-well plates overnight. They were then pre-treated with either 5 μM CA-074Me (cathepsin inhibitor) or 0.1% DMSO in complete DMEM for 1 h. The media was replaced with Advanced DMEM (Invitrogen, Carlsbad, CA, USA) supplemented with 1% FBS and 10 μM FITC-labeled PAs, and the cells were incubated for 2 h. Hoechst (5 µg/mL) was added to the media 30 min before the end of the PA incubation. The treatment media was then removed, and the cells were washed and fixed immediately with 4% paraformaldehyde in PBS for 10 min at room temperature. The fixed cells were then washed, and the coverslips were mounted on glass slides before imaging. Confocal images were collected on Leica TCS SP2 AOBS Laser Scanning Confocal microscope. The acquisition parameters, shutters, filter positions, and focus were controlled by LCS Leica confocal software LASAF 2.7.3.9723 (Leica, Wetzlar, Germany).

## 3. Results

### 3.1. Peptide Amphiphile Design

To test our PA delivery strategy for therapeutic peptides, we constructed a set of PAs to mimic the BH3 death domain of BIM (BIM_A_ BH3; Figure 1). However, this peptide sequence is sparingly soluble in water, so five C-terminal lysines were added to increase the charge and solubility of the peptide (BIM_A,K_) (Figure 1a). This peptide was attached to two different lipid-based tails, both of which our group and others have previously used to deliver peptides into cells, to form BIM_A,K_PA_1_ and BIM_A,K_PA_2_ (Figure 1b, Appendix A). The lipid tails were attached to the side chain of the C-terminal lysine of the BIM BH3 peptides. Finally, a cathepsin-cleavable linker was incorporated between the therapeutic peptide and the C-terminal lysines to form a cathepsin-cleavable PA, BIM_A,cath,K_PA_2_, which we hypothesized would allow for release of the peptide cargo following PA uptake into the cell. 

### 3.2. PAs Enhance Cellular Uptake Without Non-Specific Membrane Disruption

The ability of the PAs to deliver the BIM_A_ BH3 peptide into cells was tested next, as these lipid tails have previously been shown to facilitate cellular internalization [11,12,13]. To determine the extent of intracellular localization, HeLa cells were incubated in the presence of FITC-labeled peptides or PAs for 2 h followed by imaging using live cell confocal microscopy. Prior to imaging, the cells were washed to remove non-cell associated PAs. BIM_A,K_ peptide alone was not taken up into cells, but the addition of a diC16 tail (BIM_A,K_PA_1_) or a DSPE-PEG tail (BIM_A,K_PA_2_) enabled cell uptake of the otherwise cell-impermeable peptide (Figure 2a). The DSPE-PEG PA, BIM_A,K_PA_2_, was more localized to the cellular membrane at this time point, while the diC16 PA, BIM_A,K_PA_1_, had a more diffuse intracellular presence. Importantly, both BIM_A,K_PA_2_ and BIM_A,cath,K_PA_2_ had greater presence at the membrane and in punctate organelles when compared to BIM_A,K_PA_1_, suggesting poorer penetrating ability of the DSPE-PEG tail compared to the diC16 tail. Incorporation of a hydrophilic PEG domain has previously been shown to affect membrane interactions and uptake mechanisms [13], with a PEG spacer causing cellular internalization more dependent upon the active uptake mechanisms. Interestingly, when a cathepsin-cleavable linker was added between the peptide and DSPE-PEG tail domain in BIM_A,cath,K_PA_2_, the intracellular peptide more quickly became diffuse and co-localized with mitochondria, the site of action of the BCL-2 family of proteins (Appendix A). This localization of BIM_A,cath,K_PA_2_ was observed to be time dependent (Appendix A).

As the proposed mechanism(s) of PA uptake involves interactions of the lipid domains with the cell membrane, it is critical to test for non-specific membrane disruption and cytotoxicity caused by the lipid tails [21]. To rule out non-specific membrane disruption, the release of cytoplasmic LDH from cells treated with PAs was measured 1 h following treatment (Figure 2b). BIM_A,K_PA_1_, which readily entered the cells, caused dose-dependent LDH release, indicating some degree of non-specific lipid-associated membrane disruption. The DSPE-PEG tail of BIM_A,K_PA_2_, however, caused no measurable LDH release, with or without the cathepsin-cleavable linker (Figure 2b). Based upon its facile intracellular penetration, lack of non-specific cellular membrane disruption, and ability to diffusely disseminate the BIM BH3 peptide, BIM_A,cath,K_PA_2_ appeared to be a logical candidate PA for further structural, target binding, and efficacy testing compared to BIM_A,K_PA_2_.

### 3.3. Biophysical Characterization of Micelles

The critical micelle concentration (CMC) was 1.04 μM and 1.54 μM for BIM_A,K_PA_2_ and BIM_A,cath,K_PA_2_, respectively (Figure 3a). Dynamic light scattering (DLS) was used to measure the mean hydrodynamic radii (R_h_) of the micelles, which were 53.7 nm (±8.1 nm) for BIM_A,K_PA_2_ and 85.4 nm (±10.0 nm) for BIM_A,cath,K_PA_2_ (Figure 3b). Imaging the micelles with negative-stain TEM confirmed the micelles were spherical, as expected for DSPE-PEG micelles, and the sizes agreed with the DLS measurements (Figure 3c). Incorporation of a peptide into a micelle has been shown to increase its natural α-helical structure formation [31,32,33], so we next used circular dichroism (CD) to measure the alpha helicity of BIM_A,K_ after incorporation into micelles. BIM BH3 peptides within both PAs had similar degrees of alpha helicity, which were constant at temperatures ranging from 25 to 50 °C and after heating to 70 °C followed by cooling to 37 °C (Figure 3d). 

### 3.4. Target Protein Binding

Specific apoptosis induction within cells requires selective binding of the BIM BH3 peptide to hydrophobic grooves formed by selective helical domains of anti-apoptotic and pro-apoptotic BCL-2 family target proteins [19,21,22,34]. The addition of a large lipid tail, while facilitating intracellular delivery of BIM_A,K_PA_2_ and BIM_A,cath,K_PA_2_, may sterically inhibit the BIM BH3 peptide binding to its cognate target binding region. To test this, fluorescence polarization (FP) was used to measure the binding of FITC-labeled BIM BH3 peptides and PAs to recombinant antiapoptotic BCL-2 family proteins [19]. Indeed, the addition of a DSPE-PEG tail inhibited BIM_A,K_’s ability to bind to BCL-2, BCL-W, BCL-XL, and MCL-1 (Figure 4a). The BIM_A,K_ peptide alone bound each protein with double-digit nanomolar affinity. However, the addition of the DSPE-PEG tail decreased these affinities by ~2–5 fold (Figure 4a). Of note, BIM_A,K_ bound with affinities between 76–99 nM while BIM_A_ peptide (lacking the C-terminal lysines) is known to bind these same antiapoptotic targets with ~10× fold higher affinities, indicating that the addition of the lysines also likely dampened cognate target protein binding [19,21,35].

Based upon the intact PA:target protein binding affinities, we next aimed to determine if the release of the BIM BH3 peptide from both the C-terminal lysines and DSPE-PEG lipid tail could improve binding to BCL-2 family antiapoptotic protein targets. To do this, BIM_A,cath,K_PA_2_ was preincubated with recombinant cathepsin B enzyme prior to incubation with anti-apoptotic BCL-2 family proteins. Cathepsin pre-incubation resulted in time dependent increased affinities to BCL-XL plateauing between 15–30 min (Figure 4b). Incubation of BIM_A,cath,K_PA_2_ with recombinant cathepsin for 30 min led to increased BIM BH3 peptide affinity for BCL-XL and MCL-1 (Figure 4c). In fact, following the cleavage of the BIM BH3 native peptide from the C-terminal lysines and lipid tail, peptide affinity increased 6–10 fold compared to BIM_A,K_, presumably due to release from the C-terminal lysines (87 nM to 14 nM for BCL-XL and 99 nM to 10 nM for MCL-1) (Figure 4c). These binding profiles reflect previously published reports of affinities of the BIM_A_ peptide for these proteins [19,21]. These data suggest that following intracellular delivery of intact PAs, removing the lipid tail and C-terminal lysines from BIM_A_ is critical not only for the intracellular trafficking of the therapeutic peptide, but also for its ability to bind to its target proteins.

### 3.5. Cathepsin Dependence for Intracellular Accumulation

We next tested whether or not BIM_A,cath,K_PA_2_‘s diffuse intracellular trafficking (Figure 2a) depended on cathepsin cleavage of the BIM_A_ peptide from the DSPE-PEG tail by using cathepsin inhibitor CA-074Me. The cathepsin inhibitory effect of CA-074Me was first tested in vitro using recombinant cathepsin B added to a cathepsin-substrate linker that becomes fluorescent upon cathepsin cleavage (Appendix A). Cathepsin B caused a time-dependent increase in fluorescence while the addition of CA-074Me blocked reporter substrate cleavage. 

To determine the importance of cathepsin cleavage on intracellular localization and trafficking of the BIM_A_ peptide, WT mouse embryonic fibroblasts (MEFs) were pre-treated with CA-074Me or a DMSO control for 1 h followed by incubation with 10 μM BIM_A,cath,K_PA_2_. In contrast to the more rapid intracellular localization of FITC-BIM_A,cath,K_PA_2_ in control-treated WT MEFs, those pre-treated with CA-074Me showed FITC-BIM_A,cath,K_PA_2_ localized primarily near the cell surface after 1 h (Appendix A). However, 2 h following treatment with BIM_A,cath,K_PA_2_, FITC-BIM_A_ was located more diffusely throughout the cells (Figure 5), as was previously measured in identically treated HeLa cells (Figure 2). Interestingly, the nuclei of these cells appeared fragmented, and bright field imaging revealed anoikic, rounded cells with membrane blebbing, classical hallmarks of apoptosis (Figure 5). Meanwhile, WT MEFs pre-incubated with CA-074Me showed diminished FITC-BIM_A_ association that was confined to puncta near the edges of the cell membrane and lacked signs of apoptosis (Figure 5). These data suggest that the earlier diffuse intracellular accumulation of the FITC-BIM_A_ peptide after delivery by the cleavable DSPE-PEG tail depended to some degree on cathepsin cleavage of the linker. 

### 3.6. Apoptotic Cell Death Induction

Given there were early signs suggestive of apoptosis in MEFs treated with BIM_A,cath,K_PA_2_, we next determined if treatment with BIM_A,cath,K_PA_2_ induced dose- and time-dependent cell death in these same cells. BIM_A,cath,K_PA_2_ induced progressive dose- and time-dependent cell death and corresponding caspase-3/7 activation in WT MEFs (Figure 6a). The non-cleavable BIM_A,K_PA_2_ induced a lesser degree of cell death by 72 h with corresponding caspase activation (Figure 6b). The BIM_A,K_ peptide alone, however, which was unable to enter cells (Figure 2a), did not induce measurable cell death or caspase-3/7 activation (Figure 6c). An inert BIM BH3 peptide (BIM_B_-MRPEIWIAQELRRIGDFNAKKKKK) was used as a peptide control in these studies. Importantly, BIM_A,cath,K_PA_2_ was unable to induce caspase-3/7 activation in MEFs lacking the pro-apoptotic effector proteins BAK and BAX (BAX^-/-^BAK^-/-^ MEFs), indicating specific activation of the intrinsic pathway of apoptosis rather than non-specific mitochondrial outer membrane disruption by either the BIM_A_ peptide or the DSPE-PEG tail (Figure 6d). BIM_A,cath,K_PA_2_ and, to a lesser extent, BIM_A,K_PA_2_ also induced corresponding PARP cleavage in WT MEFs, another hallmark of apoptotic cell death (Figure 6e).

## 4. Discussion

Intracellular PPIs remain a great challenge to therapeutically target and are often therefore described as “undruggable” [36]. There is a new resurgence in research exploring how to effectively drug these challenging interfaces through orthosteric inhibition. PPIs are particularly challenging to drug using small molecules as the contact interfaces between proteins are usually distributed along geographically large surface areas and consist of a complex topographical interplay of polar and hydrophobic interactions. However, there has been some success in this area using relatively large small molecules such as the first in class BCL-2 inhibitor venetoclax, which recently gained FDA approval for use in patients with a number of hematological malignancies [37]. Peptides, on the other hand, have long been used as research tools to mimic fragments of proteins and disrupt PPIs. However, an enormous obstacle to their therapeutic relevance is their lack of intracellular accessibility. One strategy that has been quite successful at facilitating peptides’ cell uptake is hydrocarbon-stapling of naturally α-helical peptides, and multiple clinical trials are now underway using a stapled peptide that therapeutically reactivates WT p53 in cancerous cells [38,39,40]. Nanomaterials, such as PAs, are also being widely explored for building delivery vehicles to carry unmodified therapeutic peptides into cells while at the same time protecting them from extracellular degradation [14].

PAs have been used to build supramolecular nanomaterials for numerous biomedical applications including therapeutics, diagnostics, regenerative medicine, and vaccines [41]. Most of these applications involve cell-material interactions occurring at the cellular surface. However, our group and others have also found that unimer PAs can use their lipid tails to directly interact with the cell membrane and trigger cell uptake [11,12,13,17]. We have since studied the mechanisms by which a number of lipid tails facilitate such cellular penetration [13]. We have also determined that following lipid-mediated trafficking into endosomes, removal of the tail prevents lipid-mediated ejection and recycling of the peptide-laden PA back out of the cell and facilitates intracellular accumulation of the therapeutic peptide [18]. The current study expands upon these works by using cleavable PAs to deliver the bioactive BIM BH3 death domain into cells to induce apoptosis. The BIM BH3 domain is well known to potently induce apoptosis by manipulating the BCL-2 family of proteins’ PPIs through induced-fit binding of both pro- and anti-apoptotic proteins, but only when the cell membrane is chemically permeabilized [24,25]. Additionally, although stapled peptides are able to enter the cells, their potency can be greatly enhanced through outer cell membrane permeabilization [42]. 

Two similar PAs were tested, one with a PEG spacer (BIM_A,K_PA_2_) and one without (BIM_A,K_PA_1_), both with dialkyl lipid tails, for their ability to carry a BIM BH3 peptide into cells without non-specific disruption of the cell membrane. Although the diC16 PA (BIM_A,K_PA_1_) lacking the PEG spacer readily delivered the BIM_A_ peptide into cells, it also caused non-specific membrane disruption, a problematic, often not tested, off-target cytotoxic effect. Inclusion of a hydrophilic PEG spacer with a DSPE-PEG tail (BIM_A,K_PA_2_), meanwhile, eliminated membrane disruption. However, this addition led to inefficient intracellular localization of the peptide as cargo was seen primarily on or near the cell membrane surface. It is unclear if this was from tempered membrane crossing or from rapid endosomal recycling as our group has measured in other PA systems [13,18]. However, the incorporation of the endosomally-cleavable valine-citrulline cathepsin-substrate linker into the DSPE-PEG PA (BIM_A,cath,K_PA_2_) amplified the peptide’s intracellular accumulation, while still avoiding non-specific membrane disruption. It is likely the cathepsin linker allows for quick release of the BIM_A_ peptide and that the peptide was able to escape, to some extent, late endosomal/lysosomal trafficking and enter the cytoplasm. We have previously measured similar discrepancies in therapeutic peptide intracellular accumulation using a similar cleavage system between a peptide cargo and lipid tail carrier, indicating that the increased intracellular localization measured here is not a unique feature of the BIM_A_ peptide [41].

Removal of the lipid tail following cellular uptake proved to be important not only for intracellular trafficking and accumulation, but also for the peptide’s ability to bind its target protein(s) and exert its therapeutic effect. As may be expected, addition of the DSPE-PEG tail inhibited the ability of the BIM BH3 peptide to bind to all antiapoptotic proteins tested (BCL-2, BCL-W, BCL-XL, and MCL-1). While BIM_A,K_PA_2_ and BIM_A,cath,K_PA_2_ were able to bind their target proteins, they did so with 2–3 fold lower affinity compared to the parent BIM_A,K_ peptide alone. It is important to note that the concentration of PAs used in the binding assays (50 nM) was well below their critical micellar concentrations (>1 μM). Thereby, the differences in binding affinity were likely not due to supramolecular self-assembly, but rather due to steric hinderance within each unimer imparted by the DSPE-PEG tail. Cleavage of this tail, and the C-terminal lysines, greatly improved binding affinities of the BIM_A_ peptide to its target proteins, which translated to increased efficacy. 

Finally, we measured the PAs’ abilities to induce apoptosis after intracellular delivery. Both BIM_A,K_PA_2_ and BIM_A,cath,K_PA_2_ were able to enter cells and bind their target proteins, though the cathepsin-cleavable PA performed better at both cell uptake and target protein binding. It is possible that BIM_A,K_PA_2_ primarily resided in early/late endosomes near the cell surface and then rapidly recycled back to the cell membrane. While this may have also occurred to some extent with BIM_A,cath,K_PA_2_, some PA unimers may have been able to be cleaved upon transitioning to late endosomes upon acidification of these compartments and activation of cathepsin B. This transition and cleavage was largely blocked by cathepsin inhibition. As anticipated, the PAs’ abilities to induce apoptosis correlated with these prerequisites. Both PAs were able to induce apoptosis, however, the cathepsin-cleavable PA did so with faster kinetics and at lower doses. Given that the peptides were not structurally reinforced (e.g., hydrocarbon stapled), it is unclear how much of the delivered peptide was able to escape the endosomal/lysosomal network. Studies evaluating the efficiency of intact delivery of native peptides into the cytoplasm of cells are currently underway and beyond the scope of the present study.

BIM_A,cath,K_PA_2_ was designed to release the peptide from the tail in the endosome, but this study did not intentionally include any mechanisms to facilitate endosomal escape. By an unknown mechanism, the released peptide was able to escape the endosome, reach the mitochondria, and activate apoptosis. In addition to cellular uptake, endosomal escape of nanoparticles and biofunctional warheads is another monumental obstacle to intracellular delivery of biologic therapeutics [23]. More work is needed to understand how otherwise cell-impermeable peptides are able to escape the endosome following internalization. The overall secondary structure and charge of the peptide is likely to be critical to organelle escape once internalized into the cell [10,43].

This study expands on our explorations of how PA nanoparticles interact with cells and facilitate a therapeutic peptide’s intracellular uptake and dissemination. While a lipid tail can improve a peptide’s cellular uptake, our work supports the importance of removing the tail after uptake, not only to prevent recycling back out of the cell, but to prevent the tail from inhibiting the peptide’s binding to its target and thereby dampening its therapeutic efficacy. Endosomal entrapment and lack of cytoplasmic access, as well as stability in circulation (i.e., binding to serum proteins) remain formidable and poorly understood obstacles in the clinical translation of lipid-based carrier nanoparticles [14,41]. This study demonstrates, as proof-of-concept, that cathepsin-cleavable PAs can be used to deliver therapeutic peptides into cells to exert a biomedical effect. More research is warranted to understand the mechanisms behind membrane trafficking and endosomal escape in order to fully unlock the therapeutic potential of such peptide-based protein mimetics.

## Figures and Tables

**Figure 1 materials-12-02567-f001:**
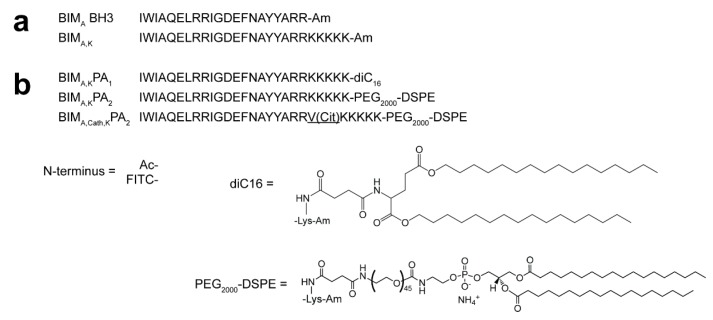
Sequences and structures used in this study. (**a**) BIM_A_ BH3 mimics the BH3 death domain of BIM. To enhance its charge and aqueous solubility, five lysines were added to the C-terminus to create BIM_A,K_. (**b**) To the BIM_A,K_ peptide, a diC16 lipid tail was added to form BIM_A,K_PA_1_, and a DSPE-PEG lipid tail was added to form BIM_A,K_PA_2_. A cathepsin-cleavable linker was also incorporated into BIM_A,K_PA_2_ between the BIM_A_ sequence and the C-terminal lysines and lipid tail to form BIM_A,cath,K_PA_2_.

**Figure 2 materials-12-02567-f002:**
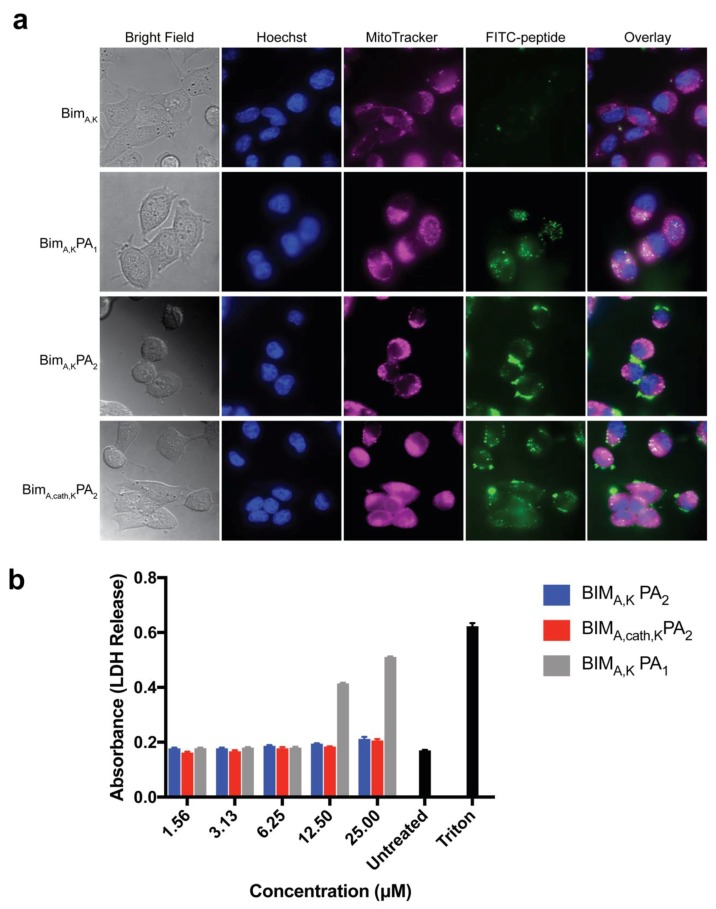
Addition of lipid tails imparts cell uptake to an otherwise cell-impermeable BIM BH3 peptide. (**a**) Live cell confocal microscopy of HeLa cells treated with fluorescein isothiocyanate (FITC)-labeled BIM_A,K_ peptide_,_ BIM_A,K_PA_1_**,** BIM_A,K_PA_2_, or BIM_A,cath,K_PA_2_ for 2 h followed by washing. BIM_A,K_PA_1_ and BIM_A,cath,K_PA_2_ resulted in diffuse intracellular localization of the BIM BH3 peptide. (**b**) BIM_A,K_PA_1_ caused non-specific membrane disruption and LDH release in a dose-dependent manner while BIM_A,K_PA_2_ and BIM_A,cath,K_PA_2_ did not. Values plotted are the mean ± S.E.M.

**Figure 3 materials-12-02567-f003:**
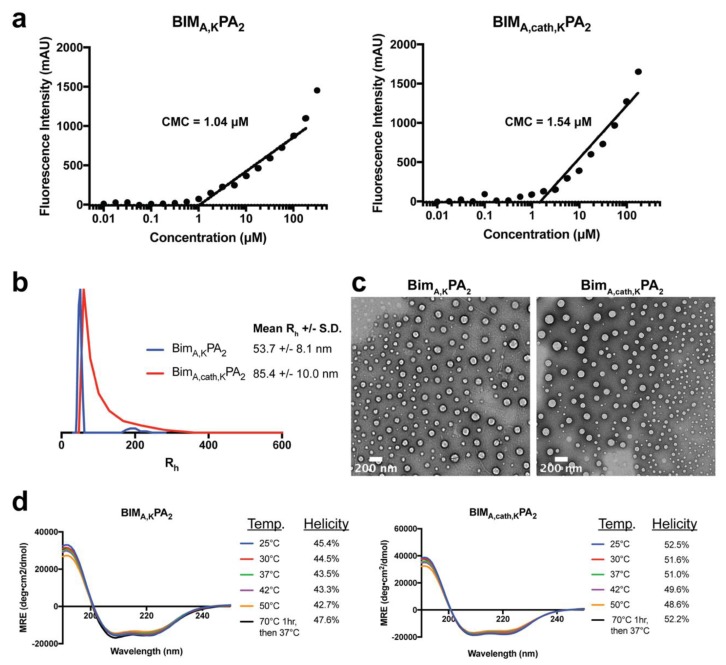
Biophysical characterization of peptide amphiphile (PA) self-assembly into micelles. (**a**) The critical micelle concentration (CMC) of BIM_A,K_PA_2_ and BIM_A,cath,K_PA_2_ as measured using 1,6-diphenyl-1,3,5-hexatriene (DPH) incorporation assay. (**b**) Dynamic light scattering (DLS) measurements of the hydrodynamic radii of BIM_A,K_PA_2_ and BIM_A,cath,K_PA_2_ indicated uniform micelle formation. (**c**) Negative-stained TEM confirmed that BIM_A,K_PA_2_ and BIM_A,cath,K_PA_2_ micelles were spherical and of equivalent sizes as measured by DLS. (**d**) Circular dichroism (CD) of the BIM BH3 peptide secondary structure within BIM_A,K_PA_2_ and BIM_A,cath,K_PA_2_ micelles revealed stable α-helicity of the BIM_A_ peptides independent of temperature.

**Figure 4 materials-12-02567-f004:**
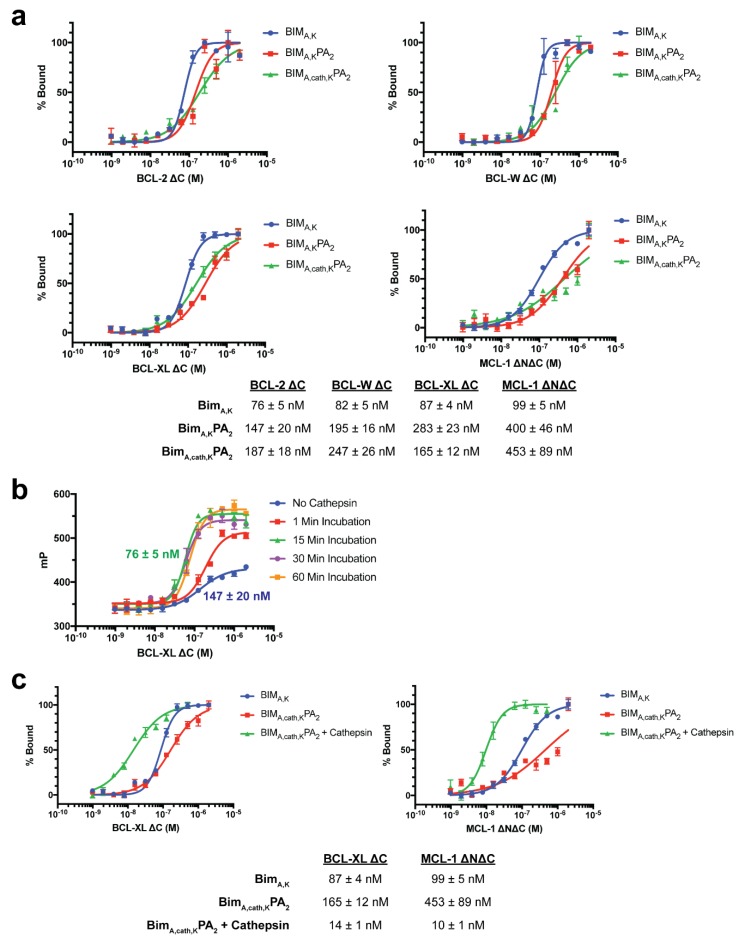
Removal of the C-terminal lipid tail and lysines of BIM_A,cath,K_PA_2_ enhances binding to antiapoptotic BCL-2 family targets. (**a**) FITC-labeled peptide and PAs were incubated with serial dilutions of recombinant BCL-2, BCL-XL, BCL-W, and MCL-1, and affinity profiles were measured by fluorescence polarization (FP). K_d_ values are listed as the mean of replicates ± S.E.M. (**b**) BIM_A,cath,K_PA_2_ pre-incubated with recombinant cathepsin B improved binding to recombinant BCL-XL in a time-dependent manner. (**c**) BIM_A,cath,K_PA_2_ pre-incubated with cathepsin B followed by enzyme-inactivation resulted in increased BIM_A_ peptide affinities to recombinant BCL-XL ΔC and MCL-1 ΔN ΔC. Plotted data and K_d_ values are the mean of replicates ± S.E.M.

**Figure 5 materials-12-02567-f005:**
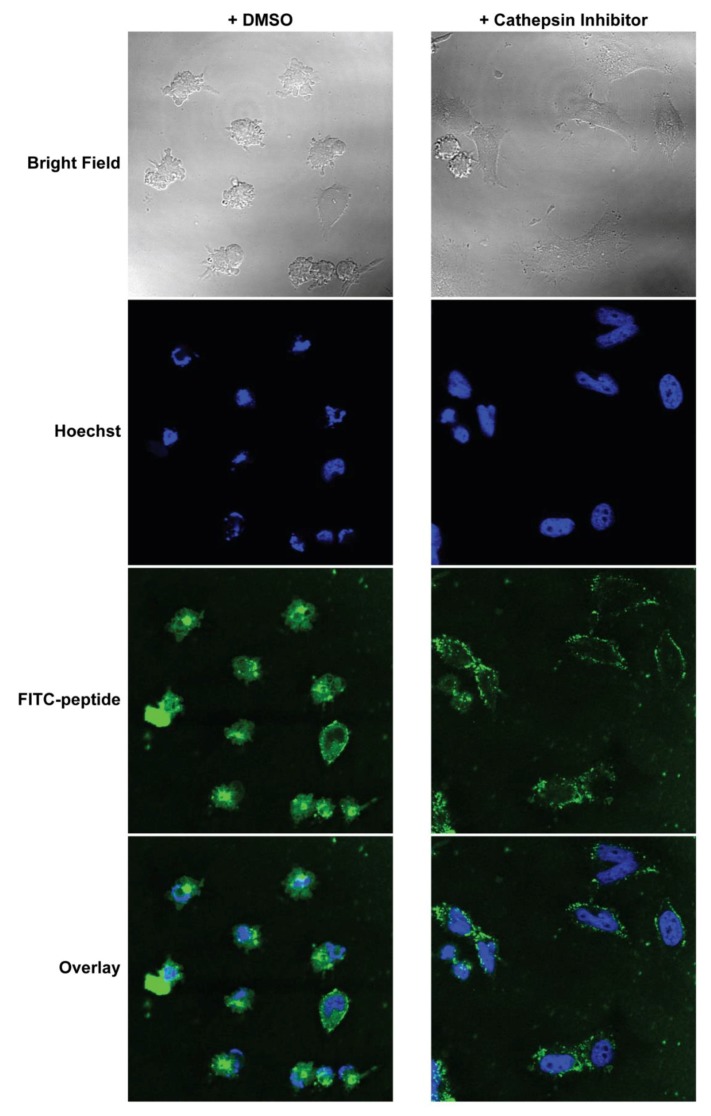
Cathepsin inhibition inhibits cellular uptake and therapeutic function of BIM_A,cath,K_PA_2_. Mouse embryonic fibroblasts (MEFs) were pre-incubated with either 5 μM CA-074Me or 0.1% (*v*/*v*) DMSO control in complete media for 1 h. The cells were then washed and treated with 10 μM FITC-BIM_A,cath,K_PA_2_ for 2 h before washing, fixation, staining with Hoechst, and confocal imaging. In the DMSO control, the FITC-BIM_A_ signal was located diffusely throughout the MEFs. Their nuclei appeared fragmented, and their cellular membranes appeared blebbed, consistent with secondary signs of apoptosis. MEFs pre-treated with CA-074Me showed FITC-BIM_A_ localized to the cell membrane and appeared otherwise morphologically normal.

**Figure 6 materials-12-02567-f006:**
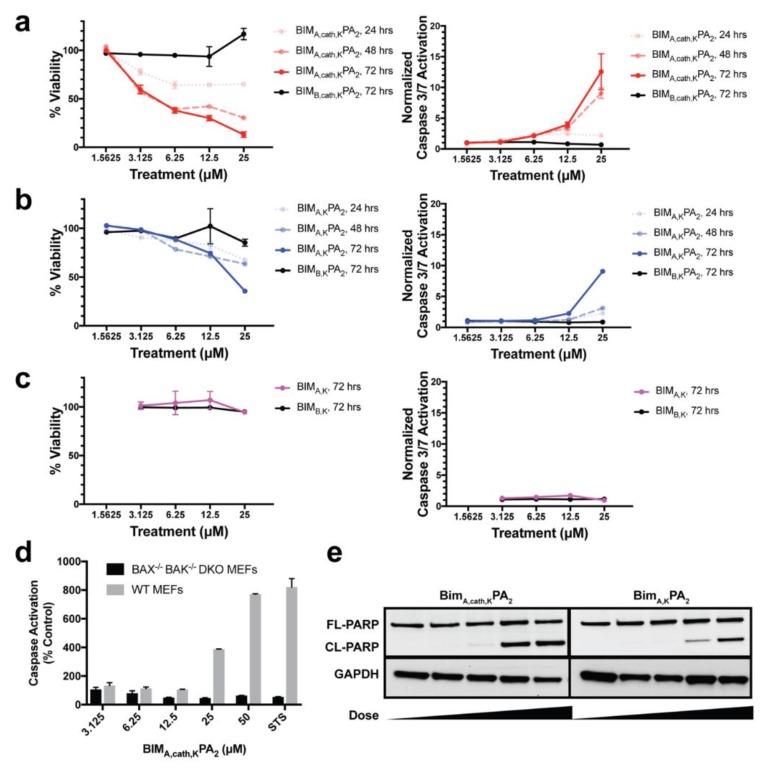
BIM_A_ PAs induce dose- and time-dependent apoptotic cell death. WT MEFs were treated with (**a**) BIM_A,cath,K_PA_2_, (**b**) BIM_A,K_PA_2_, or (**c**) BIM_A,K_ for 24, 48 and 72 h with serial dilutions of peptide or PA followed by measurement of cell viability (left column) and caspase-3/7 activation (right column). The corresponding, inactive BIM_B_ peptides were used as negative controls. The cathepsin-cleavable BIM_A,cath,K_PA_2_ induced potent cell death and corresponding caspase-3/7 activation within 24 h. The non-cleavable BIM_A,K_PA_2_ induced a lesser degree of cell death with corresponding caspase-3/7 activation by 72 h. The BIM_A,K_ peptide alone induced no measurable cell death or caspase-3/7 activation. (**d**) BIM_A,cath,K_PA_2_ induction of caspase-3/7 activation was absent in BAX^-/-^BAK^-/-^MEFs indicating on-target specificity of the BIM_A_ peptide. Staurosporine (STS; 1 μM) was used as a positive control for caspase activation. (**e**) BIM_A_ PAs induced dose-dependent PARP cleavage, another hallmark of apoptosis, as measured by Western blot analysis. FL = full length, CL = cleaved.

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
