# Peer review of "Activating the Intrinsic Pathway of Apoptosis Using BIM BH3 Peptides Delivered by Peptide Amphiphiles with Endosomal Release"

_materials, 2019, doi:10.3390/ma12162567_

Round 1

Reviewer 1 Report

Bim is a pro-apoptotic protein that plays various important roles in biology including cancer biology and immunology. Precise control for the Bim pathway by design for sure provide novel, useful therapeutics. In this manuscript, Schnorenberg et al. report new peptides BIM PAs mimicking the pro-apoptosis activity of Bim. The authors show that BIM PAs they generated show a set of advantages for the use in living cells including efficient delivery into cells. Experiments are nicely designed and well controlled: data shown in Fig.6d in which cells are unresponsive to BIM PAs when they lack BAX and BAK nicely certificate the specificity of the new peptides. I suggest that the manuscript is ready to be published at Materials as it is. As my major is biology but not chemist, I focused on biological experiments part in the manuscript.

Reviewer 2 Report

The study by Schnorenberg et al demonstrates that cathepsin B-cleavable peptide amphiphile mimicking BH3 death domain of BIM promptly enters the cytoplasm, localises to mitochondria, specifically binds Bcl-2 proteins and induces cell death. It extends the authors' previous reports on therapeutic peptide amphiphile delivery. The incorporation of the cleavable linker appears to overcome the apparent issues of reduced uptake and target binding, by liberating active peptides of interest from tagged hydrophobic tails through the endosomal cathepsin B activity. These findings imply the use of cathepsin-cleavable nanostructures as a promising deliverable peptide-based therapeutic approach.

While the study is interesting and the manuscript is well-written, I would bother the authors with a few comments on experimental design and data interpretation.

1.    Could the authors please clarify the effect of cathepsin inhibitor CA-074Me on BIMA,Cath,KPA2?

As cathepsin B is a end/lysosomal protein, where one would see the BIMA,Cath,KPAaccumulation in these compartments, upon the drug treatment. If not, it should be at least comparable to the non-cleavable BIMA,KPA2. However, as the authors described in figure 5 and figure S5, CA-074Me diminishes PA uptake and causes their accumulation on cell surface. Is CA-074Me known to block endocytosis? Is there any alternative to this drug that can be used?

 In Figure 1A, I am not convinced that BIMA,KPA2 localises to cellular membrane as it appears to concentrate in distinct spots (organelles? Golgi perhaps?).

2.    MitoTracker Red is a membrane potential dependent dye, which is not an ideal stain for mitochondria during apoptosis. It may explains why MitoTracker panels in imaging data appear diffuse. MitoTracker Green or other potential-independent dye should be used instead.

The quality of microscopy images is relatively poor. It would be great if the authors can show images with greater signals and contrast.

3.  Having all necessary experimental set-ups, I wonder why the authors did not do time-lapse live imaging. It would immensely improve the paper by addressing the kinetics of uptakes, endo/lysosomal escape and apoptosis induction.

4.    LysoTracker was mentioned in the Materials and Methods section (sub-heading 2.15), but I cannot seem to find the relevant data for it.

I hope clarification to these comments would help to improve the quality of the manuscript should it be accepted for publication.

Reviewer 3 Report

Schnorenberg and colleagues describe their work towards developing a strategy for delivering BH3 mimetics for disrupting intracellular PPIs among Bcl-2 family members. They used an established peptide amphiphile platform where they introduced a cathepsin-cleavable linker for BH3 peptide release. They design a platform that addresses important obstacles towards intracellular peptide delivery and provides a potential solution for delivering otherwise cell impermeable peptides.

A key strength of this work is the development of a delivery platform that incorporates an endosomally-cleavable linker for specific compartment based action of the two functional portions: the hydrophobic lipid tail that provides a micellar environment shielding the peptide from the extracellular milieu enhancing cellular uptake and a released free peptide cargo for functional targeting to the mitochondria. One of the weakness of this work is that that the endosomal escape mechanism of the cleaved peptide cargo is not well understood and seems to be dependent on characteristics of the specific cargo. Therefore, their PA strategy needs further evaluation to be generalizable to other possible therapeutic peptides - a point that the authors have addressed in the 'Discussion' section of the manuscript.

Important points to address:

1) In the Introduction section - lines 42-45, cancer cells upregulate anti-apoptotic members and do not downregulate pro-apoptotic members and are actually primed for apoptosis. Either a citation about their current statement should be provided or they should delete this portion and introduce the apoptotic priming mechanism of cancer cells which is why the Bcl-targeting inhibitors actually work.

2) Figure 3. Current small molecule inhibitors are effective on several cancer cell lines at concentrations less than 1μM. The fact that the PAs concentrations need to be above the CMC despite the peptide itself having a low nM affinity is a prerequisite of this delivery strategy and might be a constraint towards attaining a low therapeutic window for treatment. The authors should comment on how this limitation could be overcome.

3) Figure 4. The authors make a statement that the lipid tail inhibits binding to the anti-apopototic target alone but the EC50 data shows that cleaving the lipid tail improves the affinity by ~2 fold while cleaving the additional lysines improve it by ~6 fold. It's seems the lysines are primarily blocking the binding site thereby reducing the affinity. The authors need to modify the figure legend to include this information.

4) Materials and Methods, 2.8. - In the FP binding assay section, it is not clear why the direct binding curves were fit to EC50 values where they clearly should represent measured Kds. For comparison purposes the conclusions of the results are clear but the data need to be fit to an appropriate binding model to determine the experimental Kds.

5) Materials and Methods, 2.12. - In the cell viability assay section, it is not clear why the cells were incubated in 1% FBS for 6 hours and then the FBS concentration was increased to 10% after that. Does the albumin in FBS disrupt the DSPE-PEG micelles? If so, the authors need to comment on how this delivery platform could be improved to make it a relevant therapeutic strategy. 

Author Response

We would like to thank the reviewers for their careful and thoughtful review of our work. We have addressed the reviewers comments and hope that you will find that the alterations to the manuscript further support our conclusions. We believe these textual and graphical changes have significantly improved the manuscript. As such, we greatly appreciate your important contributions to our work.

Reviewer #3:

Schnorenberg and colleagues describe their work towards developing a strategy for delivering BH3 mimetics for disrupting intracellular PPIs among Bcl-2 family members. They used an established peptide amphiphile platform where they introduced a cathepsin-cleavable linker for BH3 peptide release. They design a platform that addresses important obstacles towards intracellular peptide delivery and provides a potential solution for delivering otherwise cell impermeable peptides.

A key strength of this work is the development of a delivery platform that incorporates an endosomally-cleavable linker for specific compartment based action of the two functional portions: the hydrophobic lipid tail that provides a micellar environment shielding the peptide from the extracellular milieu enhancing cellular uptake and a released free peptide cargo for functional targeting to the mitochondria. One of the weakness of this work is that that the endosomal escape mechanism of the cleaved peptide cargo is not well understood and seems to be dependent on characteristics of the specific cargo. Therefore, their PA strategy needs further evaluation to be generalizable to other possible therapeutic peptides - a point that the authors have addressed in the 'Discussion' section of the manuscript.

Important points to address:

Point 1: In the Introduction section - lines 42-45, cancer cells upregulate anti-apoptotic members and do not downregulate pro-apoptotic members and are actually primed for apoptosis. Either a citation about their current statement should be provided or they should delete this portion and introduce the apoptotic priming mechanism of cancer cells which is why the Bcl-targeting inhibitors actually work.

We appreciate this clarification. We have now referenced two articles that review examples of human cancers that downregulate and/or upregulate pro-apoptotic BCL-2 family members [5,6]. We have also clarified that BH3-mimetics are particularly effective “in cancers that are “primed for death” with upregulated anti-apoptotic proteins [7].”

Point 2: Figure 3. Current small molecule inhibitors are effective on several cancer cell lines at concentrations less than 1μM. The fact that the PAs concentrations need to be above the CMC despite the peptide itself having a low nM affinity is a prerequisite of this delivery strategy and might be a constraint towards attaining a low therapeutic window for treatment. The authors should comment on how this limitation could be overcome.

We appreciate the reviewer’s careful considerations of the obstacles of intracellular delivery. Indeed, the differences in peptide doses that are effective at the protein level are orders of magnitude lower than the doses that are effective at the cellular level (universally, across the literature), highlighting a therapeutically relevant dichotomy between intracellular access, for even the most promising of peptide-based drugs, and their therapeutic relevance. To get a small amount of therapeutic peptides to reach target proteins (nM dose), a large amount must often be applied at the cellular level (µM dose). We agree that this difference is an extremely important one, and it is in part, the motivation for this manuscript. In this report, we have taken a peptide with no efficacy alone at the cellular level and increased its efficacy logarithmically. However, as the reviewer points out, there is more to be done.

Importantly, we believe that comparison of peptides to small molecules’ effective doses and discussions regarding their therapeutic windows are fraught with complexities. For example:

A difference in effective doses at the protein level and at the cellular level highlight an obstacle of intracellular access. A “therapeutic window,” however, is a difference in effective doses between healthy and diseased cells. We purposefully make no mention or measurement of a therapeutic window in this manuscript for this reason. Importantly, we desired for our treatments to be above the CMC so as to help preclude free PA monomers contaminating our trials. Concentrations above the CMC are critical to use in cases of self-assembling nanoparticles to ensure uniform micelle size and comparable dosing between experiments. Perhaps a more advanced and relevant example is ALRN-6924, a first-in-class peptide therapeutic against intracellular protein-protein interactions that is currently in phase II clinical trials. Its effective doses against cancer cells are in the µM range (g. Carvajal et al., Sci. Transl. Med., 2018), yet still has a large therapeutic window.

We appreciate, as does the reviewer, the differences between the effective dose at the protein level (protein binding) and at the cellular level (cell killing). We view this difference as strong evidence of the monumental obstacles facing intracellular access of peptide-based nanoparticles and rationale for our on-going work focused on further enhancing the cellular uptake and endosomal escape of peptide-based nanotherapeutics.

Point 3: Figure 4. The authors make a statement that the lipid tail inhibits binding to the anti-apopototic target alone but the EC50 data shows that cleaving the lipid tail improves the affinity by ~2 fold while cleaving the additional lysines improve it by ~6 fold. It's seems the lysines are primarily blocking the binding site thereby reducing the affinity. The authors need to modify the figure legend to include this information.

We agree with this important point, and we have now added additional wording on the lysines’ inhibitory effect on binding, both in the main results text and in the legend of figure 4.

Point 4: Materials and Methods, 2.8. - In the FP binding assay section, it is not clear why the direct binding curves were fit to EC50 values where they clearly should represent measured Kds. For comparison purposes the conclusions of the results are clear but the data need to be fit to an appropriate binding model to determine the experimental Kds.

This is an excellent point, and we have clarified the text accordingly. These curves were indeed fitted to the experimental data using a normalized sigmoidal curve (with no EC50 or slope constraints). This is equivalent to a specific binding model with variable Hill slope, y_max constrained to 100%, and y_min constrained to 0%. In this case, the EC50 and Kdvalues are mathematically equivalent, though we agree that “Kd” is the more meaningful interpretation of these values in this context. We have changed all mentions of “EC50” to read “Kd”, and we clarified Method 2.8 to more clearly describe our data fitting process and experimental Kdcalculations.

Point 5: Materials and Methods, 2.12. - In the cell viability assay section, it is not clear why the cells were incubated in 1% FBS for 6 hours and then the FBS concentration was increased to 10% after that. Does the albumin in FBS disrupt the DSPE-PEG micelles? If so, the authors need to comment on how this delivery platform could be improved to make it a relevant therapeutic strategy. 

This is a very salient point. As a standard we have used Advanced media to treat our cells with PAs because it allows for culture of mammalian cells with reduced FBS. As the reviewer points out, proteins within FBS (e.g. albumin) have been shown to variably bind to peptides, and thus disrupt, PA monomers. This, like studies using hydrocarbon stapled peptides, induces variability between experimental compounds beyond their on-target mechanism of action. Additionally, this is a potential limitation of using spontaneously assembling lipid nanoparticles. Thereby we first incubate cells with PA’s in low serum conditions and then increase to normal serum conditions later. We have reviewed such limitations and added this to the discussion portion of the manuscript (references 14 and 41). In fact, we are currently working on making stable PAs to limit their potential disruption in serum.
